# Physiologically based cord clamping for infants ≥32$^{+0}$ weeks gestation: A randomised clinical trial and reference percentiles for heart rate and oxygen saturation for infants ≥35$^{+0}$ weeks gestation

**Shiraz Badurdeen**[1,2]*, **Peter G. Davis**[1,3,4], **Stuart B. Hooper**[2,5], **Susan Donath**[3], **Georgia A. Santomartino**[1], **Alissa Heng**[5], **Diana Zannino**[3], **Monsurul Hoq**[3], **C. Omar F Kamlin**[1], **Stefan C. Kane**[4,6], **Anthony Woodward**[6], **Calum T. Roberts**[2,7,8], **Graeme R. Polglase**[2,5], **Douglas A. Blank**[2,7,8], **on behalf of the Baby Directed Umbilical Cord Clamping (BabyDUCC) collaborative group**¶

1 Newborn Research Centre, The Royal Women's Hospital, Melbourne, Australia, 2 The Ritchie Centre, Hudson Institute of Medical Research, Melbourne, Australia, 3 Clinical Epidemiology and Biostatistics Unit and Clinical Sciences Research, Murdoch Children's Research Institute, Melbourne, Australia, 4 The University of Melbourne, Department of Obstetrics and Gynaecology, Melbourne, Australia, 5 Departments of Obstetrics and Gynaecology, Monash University, Melbourne, Australia, 6 Division of Maternity Services and Department of Maternal Fetal Medicine, The Royal Women's Hospital, Melbourne, Australia, 7 Department of Paediatrics, Monash University, Melbourne, Australia, 8 Monash Newborn, Monash Children's Hospital, Melbourne, Australia

¶ Membership of the Baby Directed Umbilical Cord Clamping (BabyDUCC) collaborative group is provided in the Acknowledgements.

* Shiraz.Badurdeen@thewomen.org.au

**Data Availability Statement:** All relevant data files are available from the Monash University Research

## Abstract

### Background

Globally, the majority of newborns requiring resuscitation at birth are full term or late-preterm infants. These infants typically have their umbilical cord clamped early (ECC) before moving to a resuscitation platform, losing the potential support of the placental circulation. Physiologically based cord clamping (PBCC) is clamping the umbilical cord after establishing lung aeration and holds promise as a readily available means of improving early newborn outcomes. In mechanically ventilated lambs, PBCC improved cardiovascular stability and reduced hypoxia. We hypothesised that PBCC compared to ECC would result in higher heart rate (HR) in infants needing resuscitation, without compromising safety.

### Methods and findings

Between 4 July 2018 and 18 May 2021, infants born at ≥32$^{+0}$ weeks' gestation with a paediatrician called to attend were enrolled in a parallel-arm randomised trial at 2 Australian perinatal centres. Following initial stimulation, infants requiring further resuscitation were randomised within 60 seconds of birth using a smartphone-accessible web link. The intervention (PBCC) was to establish lung aeration, either via positive pressure ventilation (PPV)

Repository (https://doi.org/10.26180/19579603.v1).

**Funding:** The study authors receive funding from the National Health and Medical Research Council (NH&MRC, https://www.nhmrc.gov.au/) Program Grant (#606789), Fellowships (SBH: APP545921, GRP: APP1105526, PGD: APP1059111, CTR: APP1175634), Monash University (CTR: Kathleen Tinsley Fellow and DAB: Victor Yu Fellow), and Australian Government Research Training Program Scholarships (SB and SCK). The funders had no role in study design, data collection and analysis, decision to publish, or preparation of the manuscript.

**Competing interests:** I have read the journal's policy and the authors of this manuscript have the following competing interests: PD receives salary and project support from the Australian National Health and Medical Research Council.

**Abbreviations:** bpm, beats per minute; CI, confidence interval; DCC, deferred cord clamping; ECC, early cord clamping; HR, heart rate; IQR, interquartile range; PBCC, physiologically based cord clamping; PPV, positive pressure ventilation; RCT, Randomized controlled trial; SD, standard deviation.

or effective spontaneous breathing, prior to cord clamping. The comparator was early cord clamping (ECC) prior to resuscitation. The primary outcome was mean HR between 60 to 120 seconds after birth, measured using 3-lead electrocardiogram, extracted from video recordings blinded to group allocation. Nonrandomised infants had deferred cord clamping (DCC) ≥120 seconds in the observational study arm.

Among 508 at-risk infants enrolled, 123 were randomised (*n* = 63 to PBCC, *n* = 60 to ECC). Median (interquartile range, IQR) for gestational age was 39.9 (38.3 to 40.7) weeks in PBCC infants and 39.6 (38.4 to 40.4) weeks in ECC infants. Approximately 49% and 50% of the PBCC and ECC infants were female, respectively. Five infants (PBCC = 2, ECC = 3, 4% total) had missing primary outcome data. Cord clamping occurred at a median (IQR) of 136 (126 to 150) seconds in the PBCC arm and 37 (27 to 51) seconds in the ECC arm. Mean HR between 60 to 120 seconds after birth was 154 bpm (beats per minute) for PBCC versus 158 bpm for ECC (adjusted mean difference −6 bpm, 95% confidence interval (CI) −17 to 5 bpm, *P* = 0.39). Among 31 secondary outcomes, postpartum haemorrhage ≥500 ml occurred in 34% and 32% of mothers in the PBCC and ECC arms, respectively. Two hundred ninety-five nonrandomised infants (55% female) with median (IQR) gestational age of 39.6 (38.6 to 40.6) weeks received DCC. Data from these infants was used to create percentile charts of expected HR and oxygen saturation in vigorous infants receiving DCC. The trial was limited by the small number of infants requiring prolonged or advanced resuscitation. PBCC may provide other important benefits we did not measure, including improved maternal–infant bonding and higher iron stores.

## Conclusions

In this study, we observed that PBCC resulted in similar mean HR compared to infants receiving ECC. The findings suggest that for infants ≥32$^{+0}$ weeks' gestation who receive brief, effective resuscitation at closely monitored births, PBCC does not provide additional benefit over ECC (performed after initial drying and stimulation) in terms of key physiological markers of transition. PBCC was feasible using a simple, low-cost strategy at both cesarean and vaginal births. The percentile charts of HR and oxygen saturation may guide clinicians monitoring the transition of at-risk infants who receive DCC.

## Trial registration

Australian New Zealand Clinical Trials Registry (ANZCTR) ACTRN12618000621213.

## Author summary

### Why was this study done?

- Deferred cord clamping (DCC) is beneficial for vigorous infants at birth. Keeping the umbilical cord intact for late-preterm and term infants receiving resuscitation has uncertain feasibility, safety, and efficacy.

- For infants receiving resuscitation, international guidelines recommend that respiratory support should be titrated against reference percentiles of oxygen saturation and

heart rate (HR). These percentiles were derived in the era of routine early cord clamping (ECC).

- We aimed to determine whether physiologically based cord clamping (PBCC) provides physiological benefits over ECC for infants requiring resuscitation at birth. We also aimed to update the reference percentiles of HR and oxygen saturation in vigorous infants receiving DCC.

## What did the researchers do and find?

- Based on preclinical lamb studies, we evaluated the approach of keeping the umbilical cord intact until lung aeration increases pulmonary blood flow and facilitates pulmonary gas exchange- "physiologically based cord clamping."

- Compared to standard care of ECC followed by resuscitation, there was no difference in the primary outcome of mean HR between 60 to 120 seconds after birth. There was also no difference in the secondary outcome of oxygen saturation in the 10 minutes after birth. Our findings suggest that the low-cost approach to deliver PBCC was safe and feasible, including at emergency births.

- In contrast to the 2 previous trials that had low rates of resuscitation or had a large proportion of post-randomisation exclusions, we only randomised infants assessed to require resuscitation following initial drying and stimulation after birth.

- Nonrandomised infants who were vigorous at birth had $\geq 2$ minutes of DCC at both vaginal and cesarean births. We provide updated percentiles of HR and oxygen saturation in vigorous newborns who have DCC to inform international consensus recommendations.

## What do these findings mean?

- For infants born at $\geq 32^{+0}$ weeks' gestation requiring resuscitation, this trial did not show superiority for PBCC with regard to HR between 60 to 120 seconds after birth compared with ECC.

- For infants with a significant delay in establishing pulmonary gas exchange, PBCC may yet be beneficial, and this warrants further investigation.

- Our percentile charts provide estimates of HR and oxygen saturation to guide clinicians monitoring the transition of infants who receive DCC at births routinely attended by paediatric doctors.

- Limitations include the small number of infants requiring prolonged or advanced resuscitation, reflecting the presence of close fetal monitoring and skilled newborn resuscitation providers. PBCC may provide other important benefits that we did not measure, including improved maternal–infant bonding and higher iron stores.

## Introduction

Improvements in resuscitation for near-term and term infants have the potential to reduce early newborn mortality worldwide [1,2]. For infants not breathing at birth, it is common

practice to cut the umbilical cord early and move the infant to a separate platform before commencing resuscitation. International recommendations have emphasised the importance of the early establishment of breathing, but delays are common and associated with early newborn mortality [2–4]. The removal of the placental circulation by early umbilical cord clamping, prior to the establishment of breathing, may contribute to adverse outcomes. Keeping the infant connected to the placenta until breathing is established holds promise as a low-cost and readily available means of improving newborn outcomes [2,5].

In contrast to time-based approaches, physiologically based cord clamping (PBCC) emphasises the establishment of lung aeration, pulmonary gas exchange, and the increase in pulmonary blood flow prior to cord clamping. Experiments in lambs have demonstrated that this approach takes advantage of umbilical venous return from the utero-placental circulation for cardiac preload until the pulmonary circulation is established [6]. Several recent and ongoing trials are investigating optimal cord management for preterm infants needing resuscitation [7–9]. However, studies of intact-cord resuscitation in term infants have been challenging because of randomised infants not receiving resuscitation, large proportions of post-randomisation exclusions, and exclusion of infants born by cesarean section [10,11].

Heart rate (HR) is the most reliable marker of infant wellbeing after birth. A seminal study found that preterm lambs with early cord clamping (ECC) became more bradycardic with increasing cord clamp-to-ventilation interval [12]. In this trial, we hypothesised that PBCC would result in more stable HR in infants needing resuscitation. We anticipated that this work would provide evidence to support future trials evaluating mortality and organ injury in infants with perinatal hypoxia.

Resuscitation guidelines currently recommend that infants receiving resuscitation achieve similar oxygen saturation ($SpO_2$) levels to those observed in healthy infants [3,13,14]. These targets are largely based on the percentile charts published in 2010 during the era of routine ECC [15,16]. Current practice for infants not needing resuscitation is to defer cord clamping (DCC) for >60 seconds. With this change in practice, there is a need to update the percentile charts of HR and $SpO_2$, and reevaluate the target physiological parameters prescribed for infants receiving resuscitation.

In this study we aimed:

1. To determine whether PBCC provides physiological benefits versus ECC for infants requiring resuscitation at birth.

2. To establish new normative percentile charts of HR and $SpO_2$ for healthy infants who are at risk of needing resuscitation prior to birth, but who are vigorous and receive DCC.

## Methods

### Ethics approval

This study was approved by the Human Research Ethics Committees at The Royal Women's Hospital (RWH) (reference number 17/19) and Monash Health (reference number RES-18-0000-035A).

### Study design

We conducted a 2-centre, parallel arm, superiority randomised clinical trial at the RWH and Monash Medical Centre (MMC) in Melbourne, Australia between 4 July 2018 and 18 May 2021 (ACTRN12618000621213). This study is reported as per the Consolidated Standards of Reporting Trials (CONSORT) guideline (S1 Supporting information).

## Participants

Infants born at $\geq 32^{+0}$ weeks' gestation with a request for a paediatrician to attend the birth for potential newborn compromise were eligible (including unplanned/emergency cesarean sections or cesarean sections with breech presentation, instrumental-assisted births, meconium-stained liquor, multiples, infants $< 37^{+0}$ weeks gestation, or concern for fetal distress prior to birth). Exclusion criteria were births where the maternal care team determined that DCC and postpartum oxytocin administration until $> 2$ minutes after birth were unsafe for the mother, monochorionic twins, multiples of $> 2$ fetuses, and infants with known congenital anomalies compromising cardiorespiratory transition after birth (including a major congenital heart defect and diaphragmatic hernia).

Prospective, written parental consent was sought where possible. In the event of an emergency birth, the study had approval to use written, deferred consent at the RWH study site. Circumstances that were considered appropriate for deferred consent included emergency cesarean births with evidence of fetal compromise and operative vaginal births where prospective consent was not possible.

## Randomisation and masking

We used a computer-generated randomisation sequence with random permuted block sizes of 4 or 6, generated by an independent statistician. Randomisation was stratified by study centre and indication for paediatric attendance: preterm $32^{+0}$ to $35^{+6}$ weeks' gestation, nonemergency birth $\geq 36^{+0}$ weeks' gestation, or emergency birth $\geq 36^{+0}$ weeks' gestation. Emergency births were defined as instrumental births and unplanned cesarean births.

Births were attended by both a paediatric junior doctor and a clinician-researcher (SB or DB). All infants were dried and stimulated immediately after birth. Infants were randomised if they remained apneic despite ongoing stimulation and were determined to need positive pressure ventilation (PPV) by the attending clinician. The decision to randomise an infant needed to occur within 60 seconds of birth and was performed by the researcher (assisted by a midwife at cesarean births) using a smartphone with a web link to the central Research Electronic Data Capture (REDCap) randomisation tool [17]. This method allowed expeditious group allocation, minimising delays in commencing resuscitation. Group allocation was unblinded to the clinicians and researcher present at the birth. Randomised infants received specific interventions to breathe at the discretion of the attending clinician. These included ongoing vigorous stimulation and/or respiratory support.

Infants were randomised 1:1 to either:

1. PBCC: establishment of effective pulmonary gas exchange, either via PPV or effective spontaneous breathing, prior to umbilical cord clamping, or

2. ECC: immediate cord clamping followed by resuscitation.

## Outcomes

The primary outcome was mean HR between 60 to 120 seconds after birth determined by 3-lead ECG. For each infant, the mean of the HR measurements at 10-second intervals between 60 to 120 seconds after birth was calculated, providing HR data was available at 4 data points including at least 1 data point by 80 seconds after birth. We prespecified 31 secondary outcomes as detailed in the Statistical Analysis Plan (S2 Supporting information). An additional secondary outcome was specified in the trial registration, "Rates of successful umbilical

cord blood donation as applicable." However, cord blood donation did not occur for infants recruited in the trial. Therefore, we have no data to report on this outcome.

## Procedures

With the exception of umbilical cord management, care was provided according to the Australian and New Zealand Committee on Resuscitation Neonatal Resuscitation Guidelines. Oxytocin for the prevention of postpartum haemorrhage was administered immediately after umbilical cord clamping in all study arms.

**Physiologically based cord clamping (PBCC).**   Study procedures were finalised following a feasibility study [18]. Infants received respiratory support using a mobile resuscitator (GE Healthcare, United States of America) set to pressures of 30/5 cmH$_2$O and at 0.21 FiO$_2$. A disposable colorimetric exhaled carbon dioxide detector (Pedicap, Medtronic, USA) was placed between the facemask and T-Piece. Infants in the PBCC group receiving PPV had umbilical cord clamping deferred until ≥2 minutes after birth and until ≥60 seconds after gold colour change was detected, indicating that exhaled carbon dioxide was ≥15 mm Hg and pulmonary circulation was established [19]. Infants in the PBCC group who breathed without PPV had their umbilical cord clamped at ≥2 minutes after birth.

At cesarean births, the mattress was placed in a sterile cover and rested on the mother's legs. Both the clinician and researcher scrubbed in beside the obstetric team. At vaginal births, infants randomised to PBCC were placed on a portable mattress on the end of the bed before commencing resuscitation. When the mother was in lithotomy position, the mattress was placed in a portable cot and positioned between the mother's legs [20].

**Early cord clamping (ECC).**   Infants in the ECC group had their umbilical cord clamped immediately after randomisation and were transferred to a resuscitation trolley prior to commencing respiratory support.

**Nonrandomised infants.**   Infants who were vigorous after birth were not randomised. Instead, these infants were included in the observational study arm. These infants received similar monitoring and were concurrently recruited to study healthy fetal-to-neonatal transition for infants at risk for resuscitation. At vaginal births, the infant remained on the mother's chest or abdomen. At cesarean births, the infant remained on the mattress on the mother's legs until cord clamping. Nonrandomised infants received a minimum of ≥120 seconds between birth and umbilical cord clamping. Of note, DCC at both study institutions is considered ≥60 seconds after birth in vigorous infants.

**Monitoring and data management.**   Immediately after birth, the researcher dried the infant's chest and wrist and placed 3 ECG leads and a preductal pulse oximeter. Nonsterile resuscitation and monitoring devices received post-market ethylene oxide sterilisation and were placed into premade packs. HR and SpO$_2$ were measured and displayed using a portable Intellivue X2 (Philips Healthcare, USA) or Infinity M540 (Dräger, Germany). The monitor screen and the pressure manometer were captured using a GoPro Hero Session camera (GoPro, San Mateo, California, USA), and the video was reviewed offline for data extraction. Audio recordings were used to verify events including the timing of umbilical cord clamping and time to first cry, which were called out by the researcher present at birth.

For HR (primary outcome) and SpO$_2$ levels, blinded data extraction was performed for randomised infants. The video recording was cropped to include only the monitor screen and muted to sound before being shared with an off-site researcher who was not in attendance at the birth [20]. For all infants, HR and oxygen saturation data were extracted at 10-second intervals until 10 minutes after birth. ECG and SpO$_2$ readings were only accepted if QRS complexes and plethysmograph waveforms, respectively, showed adequate signal quality.

Demographic data and clinical outcomes (until hospital discharge) that were not available at birth were extracted later from the medical record.

## Statistical analysis

**Randomised infants.** The sample size calculation was based on data from an audit of infants requiring PPV for apnoea and the previously published feasibility study [18]. We estimated that randomised infants (needing resuscitation) in the PBCC group and the ECC group would have a mean ± standard deviation (SD) HR of 140 ± 30 beats per minute (bpm) versus 120 ± 30 bpm, respectively, between 60 to 120 seconds after birth. Accepting a 2-sided alpha <0.05 and 90% power (1-beta), and to accommodate 10% attrition for the primary outcome due to monitoring failure, we planned to recruit a total sample of 120 randomised infants.

All analyses were specified a priori based on intention-to-treat. All randomised infants, regardless of exposure to the allocated treatment or adherence to the trial protocol (S3 Supporting information), were included in the analysis. The only exceptions were cases in which randomisation was performed in error prior to birth or parental consent was withdrawn, as specified in the Statistical Analysis Plan (S2 Supporting information).

For the primary outcome, linear regression was used to adjust for the randomisation strata and calculate the difference in means, 95% confidence interval (CI) and *p*-value. Subgroup analyses were performed for each stratum (excluding site). For secondary outcomes, comparison between the treatment arms was estimated using binomial regression (dichotomous outcomes), linear regression (where the summary measure is the mean), or quantile regression (where the summary measure is the median), adjusted for the stratification factors used during randomisation.

**Nonrandomised infants.** Data from nonrandomised infants who received ≥2minutes DCC per protocol and remained vigorous after cord clamping (no respiratory support) until ≥10 minutes after birth were used to develop reference percentile charts of $SpO_2$ and HR. We used methodology originally proposed by Royston and recommended by Cole for longitudinal data [21,22]. This involved fitting nonlinear regression models to the mean using fractional polynomials in minutes after birth. Mean values were estimated using mixed-effect regression using the same power variables of minutes after birth as fixed effects, as well as infant and time as random effects. Type of birth and interaction between time after birth and type of birth as covariates were included in the model based on the data and previous literature [15,16]. The percentiles were then calculated by adding or subtracting standard deviation of values multiplied by z-scores based on standard normal distribution to the mean values.

## Results

### Participants

Five hundred eight infants were enrolled (Fig 1). Two infants were excluded prior to analysis due to accidental randomisation during equipment setup and prior to birth (both allocated to ECC). After birth, 123 infants were randomised to either PBCC (*n* = 63) or ECC (*n* = 60). Among 383 nonrandomised infants, 295 were included in the observational study arm. Recruitment was concluded once the sample size for the randomised trial was reached.

The baseline characteristics reflect the high-risk population enrolled (Table 1). Aside from a larger proportion of mothers in the PBCC arm having a medical complication of pregnancy compared to the ECC arm (43% versus 25%), the groups were similar. The median gestation at birth was 39$^{+5}$ weeks in both randomised arms. Mean time to randomisation was 26 seconds after birth. Cord clamping occurred at a mean of 136 seconds and 37 seconds in the PBCC and ECC arms, respectively. Two infants in the PBCC arm (3%) had ECC due to early maternal

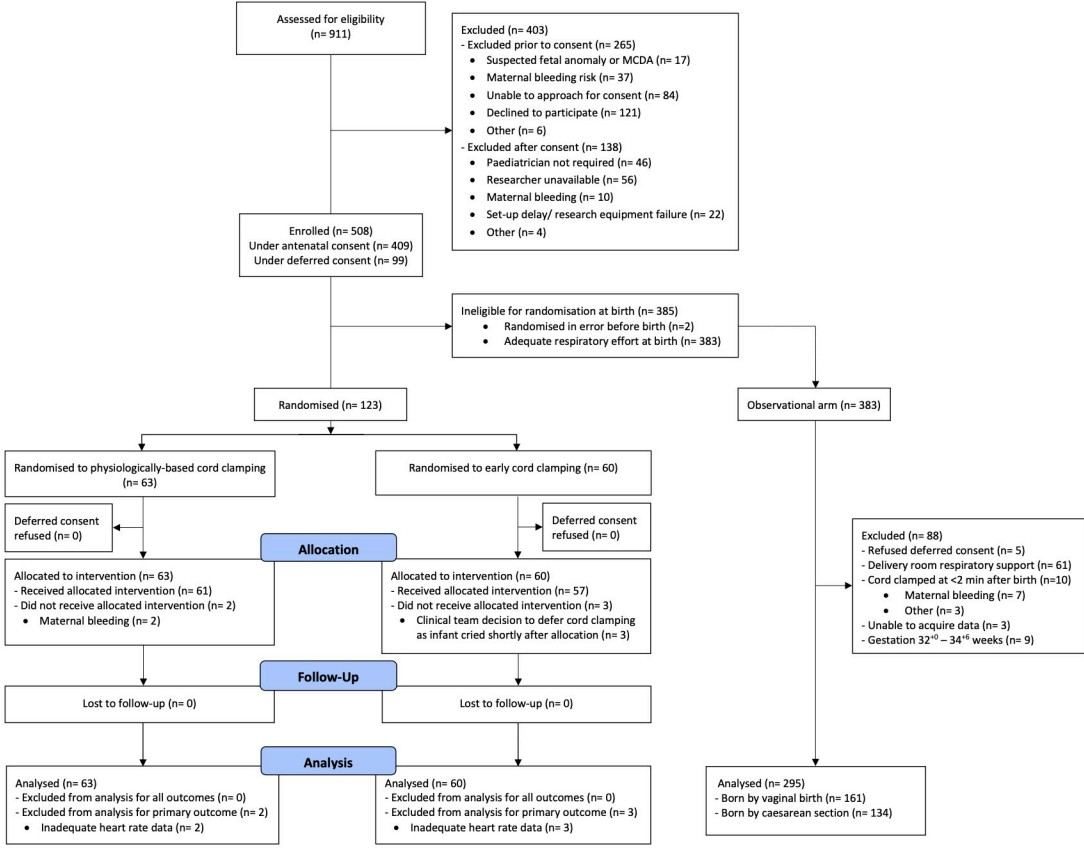

**Fig 1. Modified CONSORT diagram showing participant recruitment and allocation in the randomised trial and observational arms of the study.** MCDA, monochorionic diamniotic.

bleeding; the intervention was successfully implemented in all other infants. Three infants in the ECC arm (5%) received DCC at clinician discretion because the infants cried and had good respiratory effort immediately after randomisation.

## Primary outcome

The mean HR between 60 to 120 seconds after birth was 154 bpm (SD, 31 bpm) in the PBCC and 158 bpm (SD, 35 bpm) in the ECC arms (mean difference adjusted for randomisation strata −6 bpm, 95% CI −17 − 5 bpm, $p$-value 0.39). There was no evidence of difference between study arms in each of the subgroup analyses within the randomisation strata of preterm $32^{+0}$ to $35^{+6}$ weeks' gestation, nonemergency birth $\geq 36^{+0}$ weeks' gestation, or emergency birth $\geq 36^{+0}$ weeks' gestation (Table 2).

## Secondary outcomes

Secondary outcomes are shown in Tables 2 and 3. Following randomisation, more infants in the PBCC arm received any resuscitation (comprising stimulation to establish spontaneous breathing and/or respiratory support) compared to the ECC arm (98% versus 77%, adjusted risk difference 22.4%, 95% CI 11.3% to 34.6%). In the ECC infants, 14/60 (23%) established regular cries shortly after randomisation and did not receive resuscitation including vigorous stimulation. A higher proportion of the infants in the PBCC arm received PPV (48% versus

**Table 1. Baseline characteristics.**

| | PBCC arm (N = 63) | ECC arm (N = 60) | Observational arm (N = 295) |
|---|---|---|---|
| Hospital at birth | | | |
| - Royal Women's Hospital | 59 (94%) | 55 (92%) | 246 (83%) |
| - Monash Medical Centre | 4 (6%) | 5 (8%) | 49 (17%) |
| Antenatal consent | 51 (81%) | 45 (75%) | 245 (83%) |
| **Maternal**[*] | | | |
| Age (years), median [IQR] | 32.2 [29.4–35.2] | 32.5 [29.2–35.6] | 32.6 [29.7–35.5] |
| Primiparity | 52 (84%) | 49 (82%) | 210 (76%) |
| Any medical complication of pregnancy[†] | 27 (43%) | 15 (25%) | 77 (26%) |
| Spontaneous onset of labour | 20 (32%) | 26 (43%) | 90 (31%) |
| Antenatal oxytocin infusion | 29 (46%) | 27 (45%) | 184 (62%) |
| Strongest analgesia/anaesthesia | | | |
| - None or nitrous oxide | 3 (5%) | 8 (13%) | 25 (8%) |
| - Opiate (IV or IM) | 1 (2%) | 2 (3%) | 1 (0%) |
| - Spinal or epidural | 58 (92%) | 50 (83%) | 266 (90%) |
| - General anaesthetic | 1 (2%) | 0 (0%) | 3 (1%) |
| **Birth** | | | |
| Reason for paediatric attendance | | | |
| - Preterm <37^+0 weeks | 12 (19%) | 8 (13%) | 22 (7%) |
| - Fetal growth restriction | 4 (6%) | 1 (2%) | 14 (5%) |
| - Meconium-stained liquor | 13 (21%) | 13 (22%) | 85 (29%) |
| - Abnormal CTG | 30 (48%) | 33 (55%) | 160 (54%) |
| - Breech/transverse lie | 15 (24%) | 13 (22%) | 33 (11%) |
| - Instrumental birth | 30 (48%) | 31 (52%) | 124 (42%) |
| - Unplanned cesarean section | 16 (25%) | 13 (22%) | 92 (31%) |
| Labour complications | | | |
| - Failure to progress | 2 (3%) | 5 (8%) | 49 (17%) |
| - Prolonged second stage[#] | 14 (22%) | 18 (30%) | 73 (25%) |
| - Difficult extraction | 38 (60%) | 26 (43%) | 60 (20%) |
| - None | 15 (24%) | 17 (28%) | 119 (40%) |
| Last measured fetal heart rate (bpm), median [IQR] | 133 [110–147] | 135 [115–146] | 140 [120–150] |
| Gestational age (weeks), median [IQR] | 39.9 [38.3–40.7] | 39.6 [38.4–40.4] | 39.6 [38.6–40.6] |
| Birth weight (kg), median [IQR] | 3.42 [2.91–3.83] | 3.42 [3.00–3.80] | 3.40 [3.08–3.70] |
| Female sex | 31 (49%) | 30 (50%) | 161 (55%) |
| Time from birth (s) to | | | |
| - Randomisation, mean (SD) | 26.5 (11.7) | 26.4 (13.2) | N/A |
| - Cord clamping, median [IQR] | 136 [126–150] | 37 [27–51] | 130 [124–149] |
| - Maternal oxytocin administration, median [IQR] | 138 [129–150] | 58 [40–83] | 132 [126–150] |
| Time from birth (s) to obtain accurate data from the ECG or pulse oximeter, median [IQR] | 50 [38–63] | 48 [38–63] | 48 [38–62] |

[*]There was 1 set of twins recruited into the study with each being randomised to different arms. Characteristics of their mother are reported in each arm; thus, the mother appears in the denominator for each study arm.

[†]Includes hypertensive disorders of pregnancy, diabetes mellitus, sepsis, oligohydramnios, antepartum haemorrhage, and placenta previa.

[#]Prolonged second stage: ≥3 hours (with epidural) and ≥2 hours (without epidural) in primiparae, and ≥2 hours (with epidural) and ≥1 hour (without epidural) in multiparae.

bpm, beats per minute; CTG, cardiotocography; IQR, interquartile range; SD, standard deviation.

**Table 2. Primary outcome and secondary outcomes for continuous variables.**

| | PBCC arm N = 63 | ECC arm N = 60 | Mean difference (95% CI)* | p-value |
|---|---|---|---|---|
| **Primary outcome[†]** | | | | |
| Mean heart rate between 60 to 120 s after birth (bpm) | 154 (31) | 158 (35) | −6 (−17, 5) | 0.39[‡] |
| **Subgroup** | | | | |
| - 32+0 to 35+6 weeks' gestation | 146 (27); n = 7 | 154 (28); n = 5 | −6 (−44, 33)** | |
| - Emergency birth ≥36+0 weeks gestation | 163 (30); n = 39 | 166 (36); n = 36 | −4 (−18, 11)** | |
| - Nonemergency birth ≥36+0 weeks gestation | 135 (30); n = 15 | 142 (28); n = 16 | −8 (−30, 15)** | |
| **Secondary outcomes (continuous variables)[#]** | | | | |
| **Resuscitation** | | | | |
| Time from birth to initiate respiratory support (s)^ | 63 [44–79]; n = 40 | 93 [60–180]; n = 28 | −30 (−63, 3) | |
| Time from birth to Pedicap colour change (s)^ | 70 [50–80]; n = 27 | 74 [62–126]; n = 12 | −16 (−46, 14) | |
| Time from birth to first cry (s)^ | 57 [42–121]; n = 62 | 48 [30–68]; n = 60 | 12 (−7, 31) | |
| Infants who did not receive respiratory support: Time between randomisation and first cry (s)^ | 11 [9–18]; n = 23 | 14 [6–18]; n = 32 | −2 (−9, 5) | |
| Time to regular crying (s)^ | 74 [55–128] | 65 [48–90] | 13 (−6, 31) | |
| Maximum fraction of inspired oxygen (%) | 61 (27); n = 24 | 54 (24); n = 20 | 7 (−9, 22) | |
| Time spent with heart rate <100 bpm (s) | 12 (33) | 12 (33) | −0 (−12, 12) | |
| Time spent with heart rate >180 bpm (s) | 155 (187) | 228 (202) | −76 (−142, −10) | |
| Heart rate variability (bpm)~ | 13 (7) | 13 (8) | 0 (−3, 3) | |
| Apgar score at 1 min^ | 7 [5, 8] | 7 [6, 9] | −1.0 (−2.3, 0.3) | |
| Apgar score at 5 min^ | 9 [9, 9] | 9 [9, 9] | 0.0 (−0.0, 0.0) | |
| Apgar score at 10 min^ | 10 [9, 10]; n = 41 | 10 [10, 10]; n = 37 | 0.2 (−0.8, 1.1) | |
| First temperature (˚C) | 36.8 (0.5) | 36.9 (0.6) | −0.1 (−0.3, 0.1) | |
| Cord arterial pH | 7.2 (0.1); n = 28 | 7.2 (0.1); n = 33 | 0.0 (−0.0, 0.1) | |
| Cord arterial lactate (mmol/L) | 5.4 (2.2); n = 31 | 5.2 (1.7); n = 35 | 0.2 (−0.8, 1.1) | |
| Cord venous pH | 7.3 (0.1); n = 45 | 7.3 (0.1); n = 42 | −0.0 (−0.0, 0.0) | |
| Cord venous lactate (mmol/L) | 4.1 (1.9); n = 48 | 4.1 (1.5); n = 44 | −0.0 (−0.7, 0.7) | |
| **Infant** | | | | |
| Haematocrit level if measured, within 24 h of birth | 0.55 (0.08); n = 14 | 0.53 (0.07); n = 8 | 0.0 (−0.1, 0.1) | |
| **Maternal** | | | | |
| Maternal blood loss (ml) | 469 (326) | 518 (573) | −38 (−198, 122) | |

*Adjusted for the randomisation stratification factors.

**Adjusted for hospital site.

[†]There were 5 infants (4%) missing the primary outcome measure, thus multiple imputation was not warranted. There was no evidence of an interaction between any of the subgroups within the treatment arm.

[‡]Calculated from a linear regression model adjusted for the randomisation stratification factors.

[#]All continuous data are reported as mean (SD) unless marked with ^ where the median [IQR: Q1, Q3] is reported. Sample size is provided if it differs from the group sample size.

^Reported as median [IQR: Q1, Q3].

~Heart rate variability was defined as the standard deviation of heart rate for infants in each arm.

bpm, beats per minute; ˚C, degrees Celsius; ECC, early cord clamping; h, hours; min, minutes; mmol/L, millimoles per litre; ml, millilitres; PBCC, physiologically based cord clamping; s, seconds.

**Table 3. Secondary outcomes for categorical variables.**

| | PBCC arm N = 63 | ECC arm N = 60 | Risk difference (95% CI)** |
|---|---|---|---|
| **Resuscitation** | | | |
| Any resuscitation in the delivery room* | 62 (98%) | 46 (77%) | 22.1% (11.1%, 33.1%) |
| - Stimulation alone | 22 (35%) | 18 (30%) | |
| - Supplemental oxygen | 0 (0%) | 1 (2%) | |
| - CPAP (with or without oxygen) | 10 (16%) | 12 (20%) | |
| - Positive pressure ventilation (mask) | 30 (48%) | 15 (25%) | |
| - Intubation | 0 (0%) | 0 (0%) | |
| - Chest compressions | 0 (0%) | 0 (0%) | |
| Infants with heart rate <100 bpm (60 to 600 s) | 17 (27%) | 13 (22%) | 4.1% (−9.6%, 17.8%) |
| Infants with heart rate <100 bpm (30 to 600 s)† | 20 (32%) | 17 (28%) | 4.2% (−10.8%, 19.1%) |
| **Infant** | | | |
| Admitted for respiratory support | 8 (13%) | 5 (8%) | 6.8% (−2.2%, 15.7%) |
| Admitted for other reason | 12 (19%) | 9 (15%) | 8.4% (0.2%, 16.7%) |
| - prematurity or low birth weight alone | 5 (8%) | 5 (8%) | |
| - low glucose | 5 (8%) | 2 (3%) | |
| - other | 2 (3%) | 2 (3%) | |
| Phototherapy | 10 (16%) | 7 (12%) | 5.9% (−4.8%, 16.5%) |
| Treated for polycythaemia | 0 (0%) | 0 (0%) | |
| Exchange transfusion | 0 (0%) | 0 (0%) | |
| **Maternal** | | | |
| Postpartum haemorrhage | 21 (34%) | 19 (32%) | 3.0% (−12.6%, 18.6%) |
| - 500 ml–999 ml | 16 (26%) | 14 (23%) | |
| - ≥1,000 ml | 5 (8%) | 5 (8%) | |
| - received blood transfusion | 3 (5%) | 2 (3%) | |
| Retained placenta | 4 (6%) | 2 (3%) | 3.2% (−2.7%, 9.2%) |
| Maternal infection | 2 (3%) | 0 (0%) | |
| - following vaginal birth | 1 (3%) | 0 (0%) | |
| - up to 30 days after cesarean birth | 1 (4%) | 0 (0%) | |

*One infant (2%) in the PBCC arm and 14 (23%) in the standard care arm did not receive any resuscitation. These infants cried immediately after randomisation.

**Adjusted for the randomisation stratification factors.

†The Statistical Analysis Plan outlined that heart rate data measured 60 s to 600 s from birth will be used when analysing this outcome. Due to additional data collected from 30 s in several infants who had low heart rates, heart rate was also summarised using data collected from 30 s.

bpm, beats per minute; CPAP, continuous positive airway pressure; ECC, early cord clamping; ml, millilitres; PBCC, physiologically based cord clamping; s, seconds.

25%) and respiratory support was initiated sooner than in ECC infants (median time from birth 63 seconds versus 93 seconds, adjusted median difference −30 seconds, 95% CI −63 to 3 seconds). Median time from birth to regular cries was similar times for PBCC and ECC arms (74 seconds versus 65 seconds, adjusted median difference 13 seconds, 95% CI −6 to 31 seconds). In both arms, median Apgar score at 1 minute was 7 and improved to 9 by 5 minutes. No infants received advanced resuscitation, i.e., endotracheal intubation or chest compressions. There was no evidence of difference in the percent of infants with HR <100 bpm between PBCC and ECC arms (32% versus 28%, adjusted risk difference 4.2%, 95% CI −10.8% to 19.1%), or in the mean time spent with HR <100 bpm (12 seconds versus 12 seconds,

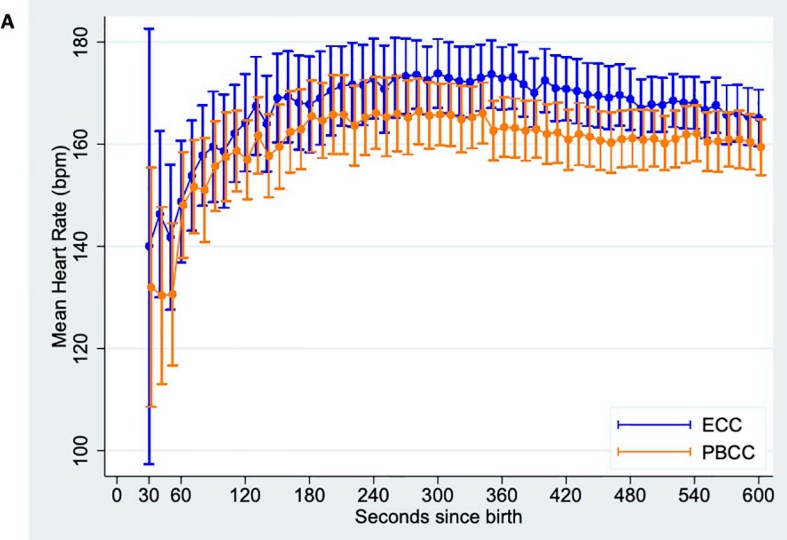

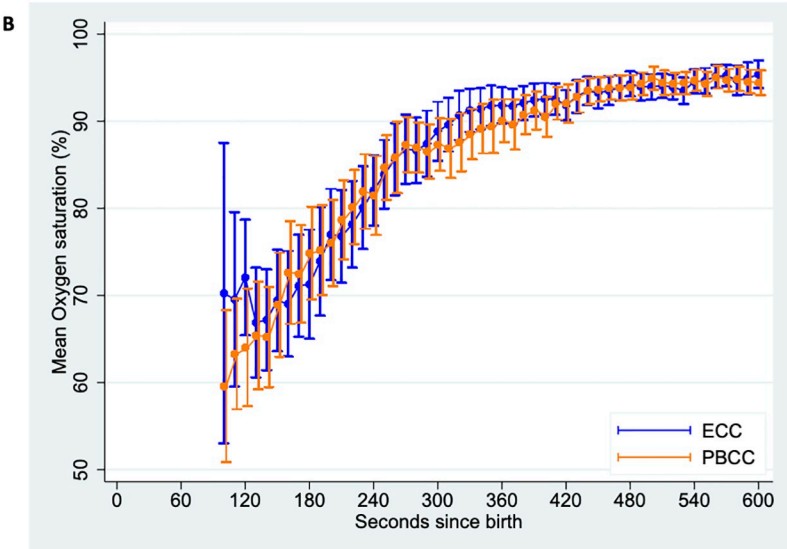

**Fig 2. Mean heart rate (A) and oxygen saturation (B) measured at 10-second intervals from 30–600 seconds from birth for infants in the PBCC and ECC arms.** Error bars represent 95% CI. CI, confidence interval; ECC, early cord clamping; PBCC, physiologically based cord clamping.

adjusted mean difference 0 second, 95% CI −12 to 12 seconds). Infants in the PBCC arm spent less time with HR >180 bpm compared to infants in the ECC arm (mean 155 seconds versus 228 seconds, adjusted mean difference −76 seconds, 95% CI −142 to −10 seconds). HR and $SpO_2$ over time were similar between study arms (Fig 2).

## Safety outcomes

There was no evidence of a difference in the percent of infants admitted to the neonatal unit for respiratory support between the PBCC and ECC arms (13% versus 8%, adjusted risk difference 6.8%, 95% CI −2.2% to 15.7%). The proportion of infants with first clinically measured

temperature <36.5 degrees Celsius was similar in the PBCC and ECC arms (21% versus 17%, risk difference 4%, 95% CI −9.8% to 17.7%). Data on other safety outcomes are shown in Table A in S4 Supporting information.

Maternal postpartum haemorrhage of ≥1 L or need for blood transfusion was similar between PBCC and ECC arms (10% versus 8%, risk difference 1.2%, 95% CI −8.9% to 11.3%). Similarly, there was no evidence of a difference in the proportion of mothers with postpartum haemorrhage between 500 ml to 999 ml without blood transfusion (PBCC = 24% versus ECC = 23%, risk difference 0.5%, 95% CI −14.5% to 15.5%).

## Observational arm

We created percentile charts using data from 295 infants born at ≥$35^{+0}$ weeks' gestation who had DCC ≥2 minutes, received no respiratory support after birth, and had adequate HR and $SpO_2$ data (Fig 1). We excluded infants born between $32^{+0}$ to $34^{+6}$ weeks' gestation because the number was small (n = 9). Among 161 vaginally born infants, 124 (77%) had instrumentally assisted births. Among 134 infants born by cesarean section, 42 (31%) had planned cesarean births, 41 (31%) had unplanned cesarean births without labour, and 51 (38%) had unplanned cesarean births in labour. First cry and umbilical cord clamping occurred at a median (interquartile range [IQR]) of 10 [3 to 36] and 130 [124 to 149] seconds, respectively. Five-minute Apgar score was a median [IQR] of 9 [9 to 9].

The progression of HR and $SpO_2$ in these infants is shown as percentile charts in Fig 3A. Infants born by cesarean birth had HR values approximately 10 bpm lower than infants born vaginally throughout the 10 minutes after birth (Fig 3B). Instances of bradycardia were uncommon: 18/295 (6%) infants had any HR <100 bpm and 53/295 (18%) had any HR <120

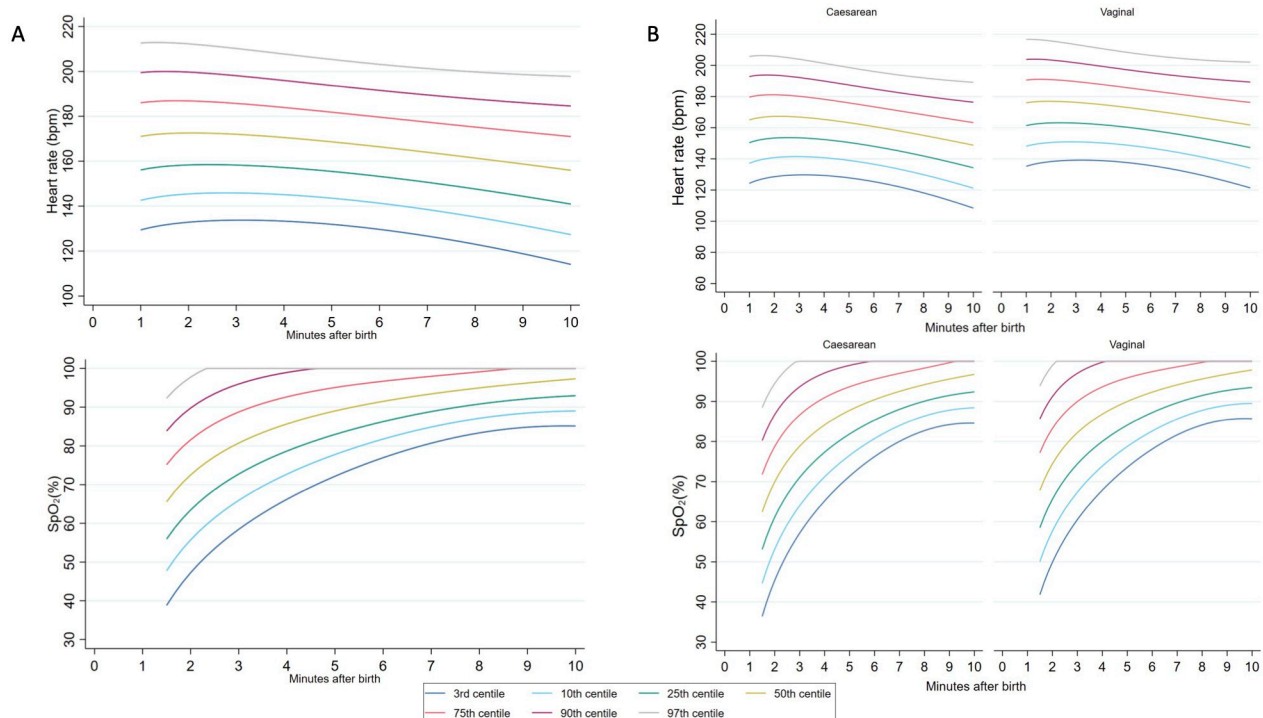

**Fig 3.** (A) Third, 10th, 25th, 50th, 75th, 90th, and 97th heart rate and preductal arterial oxygen saturation ($SpO_2$) percentiles for all infants ≥35 weeks' gestational age with no respiratory support after birth. (B) Heart rate and $SpO_2$ percentiles stratified by mode of birth.

bpm. The mean HR for infants in the randomised trial between 1 to 2 minutes after birth was similar to the 10th to 25th percentiles of infants in the observational study. HR values >180 bpm, typically considered tachycardic, were common.

Infants born by cesarean section had lower $SpO_2$ compared to vaginally born infants prior to umbilical cord clamping (Fig 3B and Tables C and D in S4 Supporting information). By 5 minutes, the differences were minimal. Approximately 6% ($n = 18$) of infants had $SpO_2$ <50% by 2 minutes. Infants in the randomised trial, many of whom received respiratory support, had $SpO_2$ values similar to the 25th percentile of infants in the observational study between 2 to 3 minutes after birth, and by 5 minutes, $SpO_2$ values were similar to the 50th percentile.

## Discussion

In this randomised trial, we found that PBCC for infants requiring resuscitation after birth resulted in similar HR and $SpO_2$ compared to infants receiving early cord clamping. PBCC was feasible and safety outcomes were similar. We also report updated percentile charts of HR and $SpO_2$ in late-preterm and term infants who received DCC and did not receive resuscitation.

Our findings indicate that PBCC does not provide additional benefit in terms of key physiological markers of transition for infants $\geq 32^{+0}$ weeks' gestation who are monitored closely and require limited resuscitation at birth, when compared to ECC that is followed by early resuscitation by skilled providers. We demonstrated feasibility of a simple, low-cost strategy of intact cord resuscitation at both cesarean and vaginal births with engagement of medical, midwifery, and theatre staff. In our small sample size, we found no difference in the risk of postpartum haemorrhage between PBCC and ECC, similar to other recent studies [23,24]. The proportion of mothers having postpartum haemorrhage in the trial was high, likely reflecting the high-risk births recruited to the trial. Mothers considered at significant risk of postpartum haemorrhage were excluded prior to enrolment at the discretion of the maternal care team because of the planned deferment of cord clamping and uterotonic administration for at least 2 minutes after birth. However, the proportion of mothers excluded from enrolment for this reason was small (37/911, 4%).

This trial provided an opportunity to concurrently study the unassisted transition of infants who were at-risk of needing resuscitation at both cesarean and vaginal births and who received DCC. Our cohort is representative of births where HR and $SpO_2$ monitoring is likely to be used, i.e., when there is fetal or anticipated neonatal compromise. It may be more appropriate for reference physiological percentiles to be derived from the population of at-risk births.

In comparison with 2 previous trials that allocated participants prior to birth and had low rates of resuscitation or a large proportion of post-randomisation exclusions [10,11], we provide a methodologically robust evaluation of the effect of PBCC on key physiological markers during the transition of late preterm and term infants. In contrast to experiments in preterm lambs, we did not observe a difference in HR or $SpO_2$ between PBCC and ECC groups. Several possible explanations exist for our findings. The number of infants who had prolonged apnoea was low because labour was closely monitored and fetuses with intra-uterine hypoxia were delivered expeditiously. Randomised infants were most likely in primary apnoea, whereby mild or transient hypoxia caused a suppression of breathing. Consequently, breathing could typically be established shortly after commencing resuscitation. Infants in the ECC group had cord clamping at a median (IQR) of 37 seconds (27 to 51 seconds) after birth, which is later than the historical approach of immediate cord clamping. This timeframe is consistent with current practice, where initial stimulation is provided with the cord intact, allowing many infants who will spontaneously cry the opportunity to have DCC. These 2 factors may have

contributed to short clamp-to-ventilation intervals and better than anticipated transition in infants receiving ECC. The approach of trying to initially stimulate spontaneous breathing prior to ECC and the need to subsequently move the infant to a separate resuscitation platform reflects current practice and explains the relatively long interval from birth to commencement of respiratory support.

A greater proportion of infants in the PBCC arm received PPV, and PPV was commenced sooner than in the ECC arm. However, the time to establish regular cries and Apgar scores were similar between groups. These findings may be due to chance or reflect physician behaviour. Respiratory support was available immediately for the PBCC infants. If randomised to the ECC arm, the physician would need to wait until the infant was moved to the resuscitation trolley, during which time we found that many infants cried, perhaps due to the extra stimulation associated with the transfer. However, this form of stimulation was outside the definition of resuscitation in the trial. The early provision of PPV in the PBCC arm may be undesirable for infants in primary apnoea who are likely to spontaneously breathe. Early provision of PPV may be helpful in infants who have been exposed to prolonged or more severe hypoxia, as it is much more difficult to stimulate spontaneous breathing in these infants. It is also possible that ECC results in acute physiological and/or hormonal changes in the infant that contribute to spontaneous breathing [25].

The HR percentiles we describe from the observational arm are markedly higher than previously reported. At 2 minutes after birth, the median HR at cesarean and vaginal births was 170 and 179 bpm, respectively, compared to 136 and 143 bpm, respectively, reported by Dawson and colleagues [16]. Recent studies in vaginally born term infants who had DCC also reported lower HR at 2 minutes [26,27]. Infants in these studies had HR measured using a Masimo pulse oximeter that has been demonstrated to underestimate early HR, possibly reflecting initial poor peripheral perfusion, in comparison to HR measured by ECG or Nellcor oximeter [28,29]. The higher HR in our cohort is therefore likely to represent both the use of ECG and elevated levels of fetal stress in the at-risk group recruited.

Infants in the randomised trial had relatively lower HR in the first 3 minutes after birth compared to most infants in the observational arm. The difference in the early HR curves likely represents the relative hypoxia of infants in the randomised arms, reflecting appropriate randomisation of infants who required resuscitation versus infants in the observational arm who were less hypoxic at birth and therefore able to transition unassisted. This observation further supports the argument for a more nuanced appreciation of HR as an early marker of successful transition beyond the traditional dichotomisation around 100 bpm [30]. HR values previously considered to be in the tachycardic range (>180 bpm) appear to be a normal physiological adaption in the first minutes after birth and may be associated with a lower risk of requiring resuscitation [30].

Resuscitation guidelines have generally used approximations of the 25th percentile of Dawson's nomogram as the lower acceptable $SpO_2$ level, broadly recommending preductal $SpO_2$ levels >60 to 65% by 2 minutes and >80% by 5 minutes [3,13,14]. Our results were consistent with these recommendations. Our $SpO_2$ percentiles likely reflect the competing influences of relative perinatal hypoxia among at-risk births attended by paediatric doctors and the positive effect of DCC on placental oxygenation and pulmonary blood flow. Padilla-Sanchez and colleagues recently reported higher $SpO_2$ values, but these were following low-risk vaginal births with DCC [26]. In addition, readings with low pulsatility index were excluded, which may have selected for infants with better peripheral perfusion and less hypoxia at birth.

The strengths of this trial include the methodological design that allowed randomisation to be rapidly performed once the infant was assessed as needing resuscitation. Over 20% of the randomised infants were enrolled via deferred consent, ensuring that we included infants with

fetal compromise significant enough to require emergency birth. The low-cost method we developed was feasible for providing respiratory support with intact cord at both cesarean and vaginal births. We were able to provide respiratory support successfully to all PBCC infants; the 2 protocol violations were due to concerns of maternal bleeding. Nearly all paediatric doctors resuscitating the infants were trainees who were new to this technique. We used the gold-standard method for ascertaining HR-ECG, with blinding for the assessment of the primary outcome (HR) and $SpO_2$. We used a video recording of the monitor to extract the HR and $SpO_2$ data rather than relying on a download of data from the monitor. Time points with poor ECG tracing or pulse oximetry waveforms were excluded. We chose this method of manual data extraction to verify good ECG and/or pulse oximetry signal, and to represent real-world interpretation of the readings. A limitation of our methodology is that we collected data at 10-second intervals rather than more frequent intervals. However, this interval captured clinically significant bradycardia or desaturation events.

For the creation of the percentile charts, we used statistical methodology that accounts for the variation in repeated measurements for each infant over time [21]. The HR and $SpO_2$ of the infants in the observational arm typically followed the initial centile line (Figs A and B in S4 Supporting information). This is informative, for instance, when monitoring infants who start with low $SpO_2$ levels but progress along the lower centiles, for whom supplemental oxygen therapy based on time-based cross-sectional thresholds may be inappropriate.

The major limitation was that we were unable to recruit a cohort of infants requiring prolonged or advanced resuscitation. Despite randomisation, a greater proportion of mothers of PBCC infants had complications of pregnancy (including hypertensive disorders of pregnancy, diabetes mellitus, sepsis, oligohydramnios, antepartum haemorrhage, and placenta previa) than mothers of ECC infants. However, there were no other baseline differences between randomised groups (Table 1). Trial participation may have improved the ability of clinicians to intervene early before infants in either arm became significantly compromised. For example, continuous HR and $SpO_2$ were established shortly after birth by the researcher instead of being the responsibility of the clinical team, leading to early awareness of compromise and early intervention. The degree of confidence on safety outcomes being similar between the randomised groups is limited by the relatively small sample size.

We did not measure other outcomes that may be of clinical relevance. Haematocrit was only measured in a small proportion of infants as clinically indicated. It is possible that the PBCC approach extends the benefits of placental transfusion/redistribution to infants requiring resuscitation who would typically have their cord clamped early [3,31]. Additionally, PBCC may promote infant–mother bonding. Our consent rate was high, and the trial received strong support by midwifery colleagues. However, we did not collect data on parental or staff perception of the study intervention.

There is a need to understand if PBCC is beneficial to infants born at the more severe end of the spectrum of perinatal hypoxia. There are published pilot trials and ongoing randomised trials investigating PBCC in preterm infants and term infants with congenital diaphragmatic hernia, but there are scarce methodologically robust data on late preterm and term infants from a variety of settings [9,32–34]. Ongoing placental circulation during PBCC to improve neonatal stability is potentially available at all births worldwide and should be evaluated in resource-limited settings. However, the reason for perinatal compromise may be due to disrupted or inadequate placental circulation, especially in term infants. In addition, other advances in resuscitation may be more impactful than PBCC. Consumers and policymakers may also interpret our findings as evidence that the known benefits of DCC (in relation to iron stores and neurodevelopmental outcomes) can be feasibly extended to infants who require resuscitation at birth [35].

Our percentile charts have implications for the target physiological parameters used for infants being supported during the perinatal transition. International recommendations currently use targets derived from low-risk infants studied during the era of ECC and should be updated considering the current standard practice of DCC [15,16,26,27]. However, a larger data set that includes infants from other centres would improve the generalisability of the data.

In summary, our results indicate that PBCC is comparable to ECC in terms of HR and $SpO_2$ for near-term and term infants mostly in primary apnoea who establish breathing soon after birth. Our findings suggest that PBCC is safe and feasible, including at emergency births. PBCC may yet be beneficial in infants who have more severe hypoxia and require more time and/or respiratory support after birth to establish pulmonary gas exchange, particularly if the utero-placental circulation is intact. This warrants further investigation alongside safety and long-term outcome data in a larger study population. Our percentile charts provide estimates of HR and $SpO_2$ to guide clinicians monitoring the transition of at-risk infants who receive DCC after birth.

## Supporting information

**S1 Supporting information. CONSORT checklist.**
(DOC)

**S2 Supporting information. Statistical analysis plan.**
(PDF)

**S3 Supporting information. Trial protocol.**
(PDF)

**S4 Supporting information. Supplementary material.** Table A. Summary statistics and selected group differences (where estimable) for safety outcomes. CI, confidence interval; ECC, early cord clamping; L, litre; PBCC, physiologically based cord clamping. Table B. Percentiles tables for heart rate and oxygen saturation for all births. Table C. Percentiles tables for heart rate and oxygen saturation for cesarean births. Table D. Percentiles tables for heart rate and oxygen saturation for vaginal births. Fig A. (A) Individual heart rate trajectories for infants with heart rate >200 beats per minute (bpm) within 2 minutes of birth and (B) infants with heart rate <100 bpm within 2 minutes of birth. Fig B. (A) Individual trajectories of oxygen saturation (SpO2) for infants with SpO2 >90% within 2 minutes of birth and (B) infants with SpO2 <50% within 2 minutes of birth.
(PDF)

## Acknowledgments

The BabyDUCC collaborative group: Jennifer Dawson, Alicia Dennis, Ryan Hodges, Sue Jacobs, Arjan te Pas, Alice Stewart, and Marta ThioLluch.

The authors acknowledge and thank the families who took part in the study and the clinical teams for their support. We thank the data safety and monitoring committee (Martin Kluckow [Chair], Michael Nicholl, and Anneke Grobler) and ethics committees.

## Author Contributions

**Conceptualization:** Shiraz Badurdeen, Peter G. Davis, Stuart B. Hooper, C. Omar F Kamlin, Stefan C. Kane, Graeme R. Polglase, Douglas A. Blank.

**Data curation:** Shiraz Badurdeen, Georgia A. Santomartino, Alissa Heng.

**Formal analysis:** Shiraz Badurdeen, Diana Zannino, Monsurul Hoq.

**Funding acquisition:** Peter G. Davis, Stuart B. Hooper, Graeme R. Polglase.

**Investigation:** Shiraz Badurdeen, Georgia A. Santomartino, Alissa Heng, Anthony Woodward, Douglas A. Blank.

**Methodology:** Shiraz Badurdeen, Peter G. Davis, Stuart B. Hooper, Susan Donath, Monsurul Hoq, C. Omar F Kamlin, Stefan C. Kane, Calum T. Roberts, Graeme R. Polglase, Douglas A. Blank.

**Project administration:** Shiraz Badurdeen, Peter G. Davis, Anthony Woodward, Calum T. Roberts, Douglas A. Blank.

**Resources:** Peter G. Davis, Stuart B. Hooper, Graeme R. Polglase, Douglas A. Blank.

**Software:** Shiraz Badurdeen, Susan Donath, Diana Zannino, Monsurul Hoq.

**Supervision:** Peter G. Davis, Stuart B. Hooper, Susan Donath, Graeme R. Polglase, Douglas A. Blank.

**Validation:** Shiraz Badurdeen, Diana Zannino, Monsurul Hoq.

**Visualization:** Diana Zannino, Monsurul Hoq.

**Writing – original draft:** Shiraz Badurdeen.

**Writing – review & editing:** Shiraz Badurdeen, Peter G. Davis, Stuart B. Hooper, Susan Donath, Georgia A. Santomartino, Alissa Heng, Monsurul Hoq, C. Omar F Kamlin, Stefan C. Kane, Anthony Woodward, Calum T. Roberts, Graeme R. Polglase, Douglas A. Blank.

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
