## [Editor Report · Decision Letter 0]

29 Oct 2021

Dear Dr Badurdeen, 

Thank you for submitting your manuscript entitled "Physiologically-based cord clamping for infants ≥ 32+0 weeks gestation - a randomised clinical trial and reference percentiles for heart rate and oxygen saturation." for consideration by PLOS Medicine.

Your manuscript has now been evaluated by the PLOS Medicine editorial staff and I am writing to let you know that we would like to send your submission out for external peer review.

Please re-submit your manuscript within two working days.

Kind regards,

Louise Gaynor-Brook, MBBS PhD

Associate Editor

PLOS Medicine

---

## [Decision Letter · Decision Letter 1]

16 Mar 2022

Dear Dr. Badurdeen,

Thank you very much for submitting your manuscript "Physiologically-based cord clamping for infants ≥ 32+0 weeks gestation - a randomised clinical trial and reference percentiles for heart rate and oxygen saturation." (PMEDICINE-D-21-04508R1) for consideration at PLOS Medicine. 

Your paper was evaluated by three independent reviewers, including a statistical reviewer, and discussed among all the editors here and with an academic editor with relevant expertise. The reviews are appended at the bottom of this email and any accompanying reviewer attachments can be seen via the link below:

[LINK]

In light of these reviews, I am afraid that we will not be able to accept the manuscript for publication in the journal in its current form, but we would like to consider a revised version that addresses the reviewers' and editors' comments. Obviously we cannot make any decision about publication until we have seen the revised manuscript and your response, and we plan to seek re-review by one or more of the reviewers. 

We expect to receive your revised manuscript by Apr 06 2022 11:59PM. Please email us (plosmedicine@plos.org) if you have any questions or concerns.

We look forward to receiving your revised manuscript. 

Sincerely,

Louise Gaynor-Brook, MBBS PhD

PLOS Medicine

plosmedicine.org

General comments:

Please include line numbers in your revised manuscript, ideally not starting from 1 with each new page.

Throughout the paper, please adapt reference call-outs to the following style: "... newborn mortality worldwide [1,2]." (noting the absence of spaces within the square brackets).

Data availability:

Please provide a web address for accessing the Monash University Research Repository.

Title: Please revise your title according to PLOS Medicine's style. Please place the study design in the subtitle (ie, after a colon). We suggest “Physiologically-based cord clamping for infants ≥ 32+0 weeks gestation: A randomised clinical trial” or similar

Abstract:

Please report your abstract according to CONSORT for abstracts, following the PLOS Medicine abstract structure (Background, Methods and Findings, Conclusions): http://www.consort-statement.org/extensions?ContentWidgetId=562

Please combine the Methods and Findings sections into one section, “Methods and Findings”.

Abstract Background: Please expand upon the context of why the study is important. 

Abstract Methods and Findings:

Please provide brief demographic details of the study population (e.g. sex, gestational age, etc)

Please specify to which study arms the five infants with missing primary outcome data belonged to.

Please define IQR and CI at first use 

Please be more specific as what ‘normative data’ is

In the last sentence of the Abstract Methods and Findings section, please describe 2-3 of the main limitations of the study's methodology."

Abstract Conclusions:

Please begin your Abstract Conclusions with "In this study, we observed ..." or similar, and summarize the main findings from your study, without overstating your conclusions. Please emphasize what is new and address the implications of your study, being careful to avoid assertions of primacy. 

Author Summary:

Under ‘Why was this study done?’, please add a final bullet point outlining your study question. 

In the final bullet point of ‘What Do These Findings Mean?’, please describe the main limitations of the study in non-technical language.

Methods:

Please add the following statement, or similar, to the Methods: "This study is reported as per the Consolidated Standards of Reporting Trials (CONSORT) guideline (S1 Checklist)." When completing the checklist, please use section and paragraph numbers, rather than page numbers which will likely no longer correspond to the appropriate sections after copy-editing.

Please refer to the supplementary file name for your statistical analysis plan.

Results: 

Under ‘Secondary outcomes’ paragraph, please be clear whether the results presented are mean or median e.g. time to initiation of respiratory support

Discussion:

Please present and organize the Discussion as follows: a short, clear summary of the article's findings; what the study adds to existing research and where and why the results may differ from previous research; strengths and limitations of the study; implications and next steps for research, clinical practice, and/or public policy; one-paragraph conclusion.

Please remove all subheadings within your Discussion e.g. Limitations and other considerations

Figures:

Please define all abbreviations used in the figure legend, including in the Supporting Information files.. 

Please consider avoiding the use of red and green together in order to make your figure more accessible to those with colour blindness.

Tables:

When a p value is given, please specify the statistical test used to determine it in the table legend.

Table 3 has been mislabelled as Table 2

Please define abbreviations used in the table legend of each table, including in the Supporting Information files.

References:

Please ensure that journal name abbreviations match those found in the National Center for Biotechnology Information (NCBI) databases (http://www.ncbi.nlm.nih.gov/nlmcatalog/journals), and are appropriately formatted and capitalised.

Please name 6 authors prior to ‘et al’, and please do not use italics for journal names. 

Please also see https://journals.plos.org/plosmedicine/s/submission-guidelines#loc-references for further details on reference formatting. 

Comments from the reviewers:

Reviewer #1: Statistical review

This paper reports a randomised controlled trial comparing two approaches to resuscitation of newborn infants, it also reports updated reference charts for heart rate and oxygen saturation in newborns.

Generally the trial was well reported. I have some minor comments.

1. Abstract: postpartum hemorrhage is one of many secondary outcomes, so it is not clear why it is highlighted in the abstract. Guidance typically is to not selectively report secondary outcomes, so I would either only include primary and safety outcome, or add more about how many secondary outcomes there were (31 on the registration page) and summary of which showed differences (e.g. 'x of the 31 secondary outcomes were statistically significant...'). 

2. Methods: I would recommend that it is made clear what the primary and secondary outcomes of the study were, perhaps in an 'Outcomes' subsection. If ones that were included in the registration and protocol are not reported here then please briefly explain why, or that they will be reported separately.

3. Statistical analysis: where 'strata' is mentioned, I would clarify this is 'randomisation strata'.

4. Statistical analysis: When binary regression is mentioned, I would clarify which type (I presume not logistic regression as risk differences are reported).

5. Statistical analysis: what outcomes (or types of outcomes) was quantile regression used for? This is explained in the SAP but I would briefly summarise it in the paper.

6. Results, page 13: one of the binary outcomes(receiving of PPV) doesn't include the estimated risk difference and CIs in the text.

7. Results, page 14: for some safety outcomes 'adjusted risk difference' is used, and for others it is 'risk difference' - is this distinction correct?

James Wason

Reviewer #2: The manuscript presents a randomized controlled trial of physiologically based cord clamping (PBCC) for moderately preterm to term infants who have risk factors identified that merit presence of a pediatrician at delivery. The study adds important information on the feasibility and safety of PBCC in this group of infants who have not been extensively studied in previous trials, but who may realize significant benefit from initiation of ventilatory support with intact placental circulation. However, the results and conclusions of the study are constrained by limitations of heart rate data collection and the setting of high-resource referral centers with close monitoring, where most fetal distress was brief and not severe.

Abstract: The abstract clearly presents the aims, methods, and results. 

What did the researchers do and find? Instead of "infants assessed to require resuscitation following initial stimulation after birth", it might be more specific to say "infants assessed to require resuscitation following initial drying", as specific stimulation to breathe later in the sequence appeared to be part of the definition of resuscitation.

What do these findings mean? The definition of "early heart rate" could be debated. Although acquisition with conventional ECG leads has been shown to be accurate, application of the leads delays the acquisition of data and important differences between groups may have been lost in the first 60 seconds.

The percentile charts provide valuable new information on an at-risk group; however, the limited number of subjects may mean that these provide early guidance on this group rather than best estimates of heart rate and oxygen saturation.

Introduction: The introduction is very clear and addresses the main points concisely.

Methods: In the description of study procedures, it would be helpful to describe the sequence of events in very concrete terms. For example, instead of "infants were stimulated immediately after birth" would it be accurate to say that "infants were dried thoroughly immediately after birth" and that provided stimulation? "If the infant was non-vigorous and assessed to require ongoing resuscitation within 60 seconds of birth" leaves the reader wondering how randomization occurred in time for early clamping. Was the need for further resuscitation determined by cry/no cry after initial drying/stimulation? Did all infants judged to need further resuscitation receive specific stimulation to breathe (apart from initial drying) before beginning PPV?

On page 9 the device used to provide respiratory support is described as a mobile resuscitator from GE Healthcare, USA, but on page 10 reference is made to the NeoPuff manometer (Fisher & Paykel).

The description of the analysis of video recording is excellent and emphasizes the clever use of technology to accomplish as much "blinding" as possible. It would be helpful to further describe the method for sampling heart rate at 10 s intervals.

Statistical analysis: What was the basis for selecting the heart rates and standard deviations for the PBCC and ECC groups? Not only were the heart rate ranges quite different from those observed, the spread between ranges was considerably less.

Results: For the primary outcome, reporting heart rate to the nearest whole number would be sufficient and more realistic.

In the discussion of secondary outcomes, it would be clearer to define what was included in "resuscitation". The initial FiO2(s) for PPV in this mixed preterm/term group might also be specified.

Was the percent of infants with HR <100 calculated on any heart rate < 100 or any heart rate sampled < 100?

Similar to heart rate, time in seconds could be reported in whole numbers.

Discussion: The discussion is very thorough and brings out excellent points regarding the relatively "late" timing of early cord clamping and providing initial stimulation with cord intact. The discussion should also address the time needed for ECG signal acquisition, as this may have obscured differences in heart rate pattern between the two groups. Studies using dry-electrode ECG acquisition of heart rate recently have reported signal acquisition at 5-10 seconds (Bush JB et al ADCFN 2021, Bjorland PA et al. ADCFN 2021). The early contours of heart rate curves also appear to be different when comparing the randomized infants to those enrolled in the observational arm.

The relatively long time from birth to establishment of ventilation in the ECC group may also be worth comment. The delay of 92 seconds is virtually identical to that reported from large observational studies in Tanzania. This emphasizes how difficult it is to initiate respiratory support quickly when umbilical cord clamping is performed before an infant is physically moved to a site for resuscitation.

Conclusion: The conclusion is appropriate, with the reservations expressed previously.

Figures and Tables: Figures and tables are valuable and complete.

References: Reference 27 spells out first names rather than surnames.

General: Minor point of usage and punctuation include:

Physiologically-based cord clamping - editorial style might omit the hyphen

Diaphragmatic herniae (page 8) - is this meant to be plural?

Reviewer #3: Physiologically based cord clamping for infants ≥32+0 weeks gestation - a randomized clinical trial in reference percentiles for heart rate and oxygen saturation

Thank you for the opportunity to review Badurdeen and colleagues' manuscript, "Physiologically-based cord clamping for infants ≥32+0 weeks gestation - a randomized clinical trial in reference percentiles for heart rate and oxygen saturation". 

The authors note that deferred cord clamping is now recommended as the standard for cord care. In addition, they note that many of these studies excluded infants who were thought to be at risk for needing resuscitation. The authors note that there may be improvement in infant transition if you avoid removal of the placental circulation by early or shortly delayed cord clamping prior to the establishment of breathing and overly early cord clamping may contribute to adverse outcomes. Keeping the infant connected to the placenta until breathing is established and holds promise as a low-cost and readily available means of improving newborn outcomes. In this trial, the authors test a physiologically based cord clamping (PBCC) procedure, which emphasizes the establishment of lung aeration, pulmonary gas exchange, and the increase in pulmonary blood flow prior to cord clamping.

In addition to exploring physiologically based cord clamping (PBCC), the authors realized that they had an exceptional opportunity to explore the transition of infants thought to be compromised but not needing resuscitation.

Current resuscitation guidelines recommend that infants receiving resuscitation achieve similar oxygen saturation levels to those observed in healthy infants. These targets are largely based on percentile charts published by some of these investigators in 2010 during the era of early cord clamping. The authors suggest a need to reevaluate the target physiologic parameters prescribed for infants receiving resuscitation. 

For that reason, the authors sought two study goals: 1) to determine whether PBCC provides physiologic benefits versus ECC for infants requiring resuscitation at birth; and 2) to establish new normative percentile charts of heart rate and oxygen saturation for healthy infants who are at need of resuscitation prior to birth but who are vigorous and received deferred cord clamping. 

This study is a two center, parallel arm, superiority randomized trial conducted in two Australian centers. Participants include infants born at ≥32 weeks gestation where a pediatrician has been requested to attend birth for the potential newborn compromise. This inclusion criteria is hard to understand in a more generalizable sense and should be specifically discussed either in the text or supplemental materials. Exclusion criteria are clearly discussed and are sensible regarding the possible compromise caused by undue delay in cord clamping. 

Of note, the study had a unique enrollment and randomization procedure, one which is difficult to perform but excellently executed by this group. They received prospective written parental consent, which was deferred until after birth, when they randomized infants at <30 seconds of age if they required resuscitation. Randomization was stratified by study center and indication for pediatric attendance, including preterm birth, non-emergent birth ≥36 weeks gestation, and emergency birth ≥36 weeks gestation. Physician attendance was required for entry into the study. Infants were randomized either to physiologically based cord clamping, the establishment of an effective pulmonary gas exchange, either via positive pressure ventilation or effective, spontaneous breathing prior to cord clamping, or ECC immediate cord clamping, followed by resuscitation. The definition of early cord clamping is somewhat confusion as this group had at least a 30 second delay, based on the randomization procedure itself. 

A significant number of patients did not require resuscitation and were followed in an observational cohort. Data from these non-randomized infants who received ≥2 minutes of delayed cord clamping, per protocol, and remained vigorous after cord clamping until ≥10 minutes after birth were used to develop percentile ranges of oxygen saturation and heart rate. 

508 infants were enrolled. Aside from a larger proportion of mothers in the physiologically based cord clamping arm having medical complications in pregnancy compared to the early cord clamping arm (43% to 25%), the groups were similar. Median gestation at birth was 39+5 weeks, mean time to randomization was 26 seconds after birth, and cord clamping occurred at a mean of 136 seconds in the PBCC group and 37 seconds in the ECC arms. Ultimately, the investigators noted no difference in heart rate overall or in the sub-groups based on gestational age and indication for delivery. More infants in the PBCC arm received resuscitation compared to the ECC arm (98% versus 77%, adjusted difference 22.4%, 95% CI, 11.3% to 34.6). A higher proportion of infants in the PBCC arm received positive pressure ventilation (48% to 25%) and respiratory support was initiated sooner than in the ECC infants. No differences in safety outcomes were reported. Routine hematocrits were not done and is a small deficiency of this study. 

The authors go on to describe the observational arm and the creation of percentile charts of heart rate and oxygen saturation. In the discussion, the authors state that physiologically based cord clamping led to similar results in heart rate and saturation compared to infants receiving early cord clamping. However, it seems naïve to state that safety outcomes and clinical outcomes were similar, as noted above. The authors include a lengthy discussion around creating the new percentile charts. 

This small but well-executed study is somewhat difficult to read. The authors combine two different exercises, both with very different methodologies. Some thought should be given to splitting out these two exercises (clearly the data from the observational trial being less important but of real interest). This would allow for much of the clinical data that is included in the first supplement to be added into an article and for authors to be able to look at many of the secondary outcomes that are associated with timing of cord clamping directly in manuscript itself. In addition, if anything, the study does not support the conclusion that physiologically based cord clamping results in similar effects. That may be true of the primary outcome (mean heart rate), and there seems to be no benefit and some potential harm from the delays in physiologically based cord clamping. More caution should be noted in the authors observations and conclusions.

[LINK]

---

## [Decision Letter · Decision Letter 2]

18 May 2022

Dear Dr. Badurdeen,

Thank you very much for re-submitting your manuscript "Physiologically-based cord clamping for infants ≥ 32+0 weeks gestation: a randomised clinical trial and reference percentiles for heart rate and oxygen saturation." (PMEDICINE-D-21-04508R2) for review by PLOS Medicine.

I have discussed the paper with my colleagues and the academic editor and it was also seen again by two reviewers. I am pleased to say that provided the remaining editorial and production issues are dealt with we are planning to accept the paper for publication in the journal.

[LINK]

We expect to receive your revised manuscript within 2 working days. Please email us (plosmedicine@plos.org) if you have any questions or concerns.

We look forward to receiving the revised manuscript by May 20 2022 11:59PM.   

Sincerely,

Louise Gaynor-Brook, MBBS PhD

Senior Editor 

PLOS Medicine

plosmedicine.org

Requests from Editors:

General comments:

Please refer to specific supplementary files e.g. S1 Table in the main text of your manuscript, rather than e.g. “Supplemental file”, “Supplemental Figures”, etc 

Title: Please revise your title to specify who the reference percentiles apply to e.g. “...reference percentiles for heart rate and oxygen saturation in infants ≥35+0 weeks’ gestation”

Abstract Background: Please remove the sentence beginning “For infants not receiving resuscitation…”

Abstract Methods and Findings:

Please relocate sentences beginning “The trial was limited by the small number…” and “PBCC may provide other important…” to the end of the Methods and Findings section. 

Line 68 - Please revise to “with median (IQR) gestational age of…”

Author Summary:

Please define PBCC at first use in your Author Summary

Given the concerns of Reviewer 3 regarding potential harms, please temper assertions that “PBCC was safe and feasible” e.g. “Our findings suggest that PBCC is safe and feasible…”

Methods:

Please provide the names of the institutional review boards that provided ethical approval and reference numbers in your Methods section (this can be moved from line 684).

Please add the following statement, or similar, to the Methods: "This study is reported as per the Consolidated Standards of Reporting Trials (CONSORT) guideline (S1 Checklist)." 

For the avoidance of issues relating to consent and copyright, it would be preferable if the accompanying videos could be uploaded to the Monash University Research Repository and cited, rather than uploaded as a supplementary file 

Discussion:

Given the concerns of Reviewer 3 regarding potential harms, please temper assertions that “PBCC was safe and feasible” e.g. “Our findings suggest that PBCC is safe and feasible…”

Figures:

Please define the abbreviations in Figure 1 e.g. MCDA

References:

Please ensure that journal name abbreviations match those found in the National Center for Biotechnology Information (NCBI) databases (http://www.ncbi.nlm.nih.gov/nlmcatalog/journals), and are appropriately formatted and capitalised. E.g ref 1 should be ‘Lancet’ rather than ‘The Lancet. Lancet Publishing Group’; ref 17 should be ‘J Biomed Inform’ instead of ‘Journal of Biomedical Informatics. Academic Press Inc.’, etc 

Supplementary files: 

Please name individual supplementary files to contain an "S" and number, as per https://journals.plos.org/plosmedicine/s/supporting-information

Comments from Reviewers:

Reviewer #1: Thank you to the authors for addressing my previous comments well. I have no further issues to raise.

Reviewer #2: Thanks to the authors for the many improvements in response to reviewer comments; these have strengthened the manuscript significantly. With very minor adjustments to make language consistent through the manuscript, the revision is acceptable for publication. 

The explanation of the approach to initial care at birth prior to randomization (drying and stimulation) is a significant part of the trial that deserves emphasis in the abstract and conclusion. Although many centers now are taking this approach, it differs from the original definition of early cord clamping, which often occurred within 10-15 seconds after birth. 

Wording of the conclusion in the abstract differs slightly from that in the body of the manuscript ("limited benefits" in abstract vs."does not provide additional benefit" in line 454). The latter wording most accurately reflects the data, but the conditions of ECC performed after initial drying and stimulation deserve mention in both places. (The findings suggest that for infants>=32+0 weeks gestation who respond to brief, effective resuscitation at closely monitored births, PBCC does not demonstrate benefit over ECC performed after initial drying and stimulation in terms of key physiological markers of transition.)

If post-partum hemorrhage was the primary safety outcome, it may be pertinent to describe it as such in the abstract (e.g. Among 31 secondary outcomes, the safety outcome of post-partum haemorrhage......line 63).

In line 268, the concentration of oxygen should be expressed as a fraction (0.21 FiO2) or a percentage (21% oxygen).

The definition of deferred clamping in the introduction (line 187) as >30-60 seconds creates some confusion with ECC as practiced in the study (27-51 seconds) and the standard of care for deferred clamping at the study sites (> 60 seconds).

[LINK]

---

## [Editor Report · Decision Letter 3]

25 May 2022

Dear Dr Badurdeen, 

On behalf of my colleagues and the Academic Editor, Dr Zulfiqar A. Bhutta, I am pleased to inform you that we have agreed to publish your manuscript "Physiologically-based cord clamping for infants ≥ 32+0 weeks gestation: a randomised clinical trial and reference percentiles for heart rate and oxygen saturation for infants ≥ 35+0 weeks gestation." (PMEDICINE-D-21-04508R3) in PLOS Medicine.

PRESS

Sincerely, 

Callam Davidson

(On behalf of Louise Gaynor-Brook, MBBS PhD )

Associate Editor 

PLOS Medicine